# A Systematic Study of the Temperature Dependence of the Dielectric Function of GaSe Uniaxial Crystals from 27 to 300 K

**DOI:** 10.3390/nano14100839

**Published:** 2024-05-10

**Authors:** Long V. Le, Tien-Thanh Nguyen, Xuan Au Nguyen, Do Duc Cuong, Thi Huong Nguyen, Van Quang Nguyen, Sunglae Cho, Young Dong Kim, Tae Jung Kim

**Affiliations:** 1Institute of Materials Science, Vietnam Academy of Science and Technology, Hanoi 100000, Vietnam; ntthanh@ims.vast.ac.vn; 2Department of Physics, Kyung Hee University, Seoul 02447, Republic of Korea; xuanau@khu.ac.kr (X.A.N.); ydkim@khu.ac.kr (Y.D.K.); 3Faculty of Physics and Engineering Physics, University of Science, VNU-HCM, Ho Chi Minh City 700000, Vietnam; ddcuong@hcmus.edu.vn; 4Department of Physics, Nha Trang University, Nha Trang 650000, Vietnam; huongnt@ntu.edu.vn; 5Department of Physics and Energy Harvest Storage Research Center, University of Ulsan, Ulsan 44610, Republic of Korea; quang3012@ulsan.ac.kr (V.Q.N.); slcho@ulsan.ac.kr (S.C.); 6Advanced Process Development Team, ISAC Research Inc., Techno2ro 340, Tabrip-dong, Yuseong-gu, Daejeon 34036, Republic of Korea

**Keywords:** uniaxual crystal GaSe, spectroscopic ellipsometry, dielectric function, exciton, first-principles calculations

## Abstract

We report the temperature dependences of the dielectric function *ε* = *ε*_1_ + i*ε*_2_ and critical point (CP) energies of the uniaxial crystal GaSe in the spectral energy region from 0.74 to 6.42 eV and at temperatures from 27 to 300 K using spectroscopic ellipsometry. The fundamental bandgap and strong exciton effect near 2.1 eV are detected only in the c-direction, which is perpendicular to the cleavage plane of the crystal. The temperature dependences of the CP energies were determined by fitting the data to the phenomenological expression that incorporates the Bose–Einstein statistical factor and the temperature coefficient to describe the electron–phonon interaction. To determine the origin of this anisotropy, we perform first-principles calculations using the mBJ method for bandgap correction. The results clearly demonstrate that the anisotropic dielectric characteristics can be directly attributed to the inherent anisotropy of *p* orbitals. More specifically, this prominent excitonic feature and fundamental bandgap are derived from the band-to-band transition between *s* and *p_z_* orbitals at the Γ-point.

## 1. Introduction

Gallium selenide (GaSe) is a compound in the III–VI group that has gained considerable interest recently due to its exceptional properties and potential applications in various fields including photodetectors [1,2,3,4,5,6], water splitting [6,7,8], lasers [9,10,11,12], and nonlinear optics [13,14,15]. GaSe can crystallize in four different polytypes, namely β-, ε-, γ-, and δ-phase structures, which correspond to the space groups $P6_{3}/mmc\;\;(D_{6h}^{4})$, $P\bar{6}m2(D_{3h}^{1})$, $R3m\;\;(C_{3v}^{5})$, and $P6_{3}mc\;\;(C_{6v}^{4})$, respectively. These structures are characterized by hexagonal-layer stacking sequences that consist of two to four layers. As a result, GaSe exhibits highly anisotropic properties perpendicular to the cleavage planes, which are held together by van der Waals forces.

Knowledge of the optical properties of materials over a wide photon energy range is important for designing photonic and photovoltaic devices as well as for verifying the predictions made by the calculations of the electronic energy band structure. Numerous studies have reported on the optical properties of this material using various methods, such as Raman scattering [16,17,18,19], photoluminescence (PL) [20,21,22,23,24], absorption [25,26,27], and spectroscopic ellipsometry [28,29,30]. However, most of these studies have focused on the isotropic characteristics of the material on the cleavage plane, while the anisotropic properties are crucial for nonlinear applications. In 1971, Bourdon and Khelladi [31] reported the absorption spectra of a GaSe cleavage sample under oblique incidence. They investigated the polarization of the incident light, whether it was parallel or perpendicular to the incidence plane. Their findings revealed that the transmission coefficient, $T_{∥}$, was greater than four times that of $T_{⊥}$ in the energy range above the absorption edge at 13 K and the same behavior and the same values of $T_{∥}/T_{⊥}$ at 77 and 300 K. In 1979, the systematic study on temperature dependence of optical absorption for GaSe (on cleavage plane) near the fundamental band edge was performed firstly by Antonioli et al. [32] in the temperature range from 65 K to room temperature. In 2008, Cui et al. [33] reported the photoluminescence (PL) of GaSe and GaSe:In at 9 K. They discovered that the peak of the exciton bound to the acceptor disappeared and the peak of the donor–acceptor pair appeared in the GaSe crystal after indium doping. Zhang et al. [34] studied the temperature-dependent PL emission from unstrained and strained GaSe nanosheets and found that the formation of new peaks in strained samples can be attributed to the recombination of bound excitons. In 2022, Usman et al. [23] investigated the thickness and temperature dependencies of PL in a few-layer GaSe. They observed that the PL intensity linearly increases with the number of layers, while the peak position increases as the layer number decreases.

Spectroscopic ellipsometry (SE) is an optical technique that accurately and sensitively investigates the dielectric properties of materials. Unlike other methods, SE does not rely on Kramers–Kronig analysis [35]. It can determine both the real and imaginary parts of the dielectric function (*ε*) simultaneously. In 1973, Meyer et al. [30] conducted the first measurement of GaSe using SE at room temperature. The study aimed to determine the ordinary and extraordinary optical indices within the range of 220 (5.6 eV) to 800 nm (1.55 eV). However, it is important to note that this measurement did not include the majority of critical point (CP) energies. More recently, Choi et al. [29] and Isik et al. [36] determined CP energies on the cleavage plane of GaSe over a wider range.

Here, we provide data on the dielectric functions along the principal axes of uniaxial crystal GaSe. The data cover the spectral range of 0.74 to 6.42 eV and includes temperatures ranging from 27 to 300 K. The sample was prepared using the temperature gradient method at 980 °C, as explained in detail below. The SE measurements were conducted under ultra-high vacuum conditions. The measured pseudodielectric function data were processed to eliminate the impact of surface roughness, resulting in the bulk *ε*_a_ and *ε*_c_ values. Along the c-axis, band-to-band transitions at the Γ-point revealed a fundamental bandgap and an excitonic feature around 2.1 eV. This feature primarily stems from *s* and *p_z_* orbitals. In the a(b)-axis, the first critical point, observed at 3.36 eV (at 27 K), is attributed to band-to-band transitions at the Γ-point involving *s* and *p_x_* (*p_y_*) orbitals. To determine the critical point (CP) energies, we utilized the second derivative function of *ε* with standard analytic expressions. This information will be valuable for device engineering and enhancing our understanding of the fundamental optical properties of GaSe.

## 2. Materials and Method

### 2.1. Sample Growth and Preparation

The temperature gradient method was used to grow GaSe single crystals. The process involved preparing high-purity (99.999%) Ga and Se powders in a stoichiometric ratio for growth. The mixture was then loaded into cylindrical quartz tubes with a conical bottom and evacuated to an atmosphere of 10^−4^ Torr before being sealed using an oxygen–hydrogen flame. To protect the inner tube from breakage due to the high vapor pressure of the materials and the different thermal expansion coefficients between the samples and the quartz ampoule, another quartz tube was sealed outside. The ampoules were then placed into a vertical furnace and gradually heated to 980 °C, which is approximately 20 °C above the melting point of GaSe. They were maintained at this temperature for 16 *h* to prepare the compounds. After this, the molten material was cooled down at a very low rate of about 1 °C per hour, below the melting point.

### 2.2. Band Structure Calculation

GaSe exists in multiple polytypes, such as β, ε, γ, and δ. Our theoretical calculations specifically focused on the simplest structure, β-GaSe. Previous studies have examined the energy band calculations for both β-GaSe [37,38] and ε-GaSe [39,40,41]. Since the β and ε polytypes have nearly identical crystal structures, it is expected that their electronic band structures and optical properties would be similar [39]. The GaSe layers consist of two planes of Ga atoms sandwiched between two planes of Se atoms, as shown in Figure 1a. In each plane, Ga and Se atoms arrange themselves in a two-dimensional hexagonal lattice. The anisotropy of the $P6_{3}/mmc\;\;(D_{6h}^{4})$ structure is clearly demonstrated by the Brillouin zone (BZ) shown in Figure 1b.

First-principle density functional theory (DFT) calculations were conducted using the projector-augmented-wave formalism [42], which was implemented in the Vienna ab initio simulation package (VASP v. 5.2) [43]. The experimental determination of the structural parameters for GaSe was reported in Ref. [44]. The exchange correlation functional was described using the generalized-gradient-approximation (GGA) with Perdew, Burke, and Ernzerhof (PBE) parameterization [45,46]. It is important to note that the GGA method consistently underestimates the band gap. To obtain more accurate band gap values, the modified Becke-Johnson (mBJ) exchange potential was used in conjunction with L(S)DA correlation [47,48]. The mBJ method offers the advantage of providing band gaps that are comparable in accuracy to those obtained using hybrid functional or GW methods, while also being computationally less expensive, similar to standard DFT calculations. The plane-wave kinetic energy cutoff was set at 350 eV and the BZ integrations were performed with a 8 × 8 × 3 Γ-centered *k*-points grid.

Since the spin-orbit effects are expected to be insignificant for III–VI compounds [49,50,51], we have excluded the spin-orbit interaction from our calculations.

### 2.3. Characterization

The sample structure was characterized using X-ray diffraction (XRD) with the XD8 Advance Bucker instrument and Cu-Kα radiation. XRD data were collected in the 2θ range from 10 to 70 degrees with a scanning rate of $2.4^{o}/\min$. Raman scattering measurements were recorded using Raman spectroscopy (XploRA PLUS, Horiba, Kyoto, Japan) with an excitation wavelength of 532 nm, a neutral-density filter of 1%, and a 2400 grooves/mm grating. A 100× objective lens (N.A. = 0.8) was used to focus the laser to a spot of approximately 1 μm diameter and to collect the scattered light from the sample.

For morphological analysis, high-resolution transmission electron microscopy (HRTEM), selected-area electron diffraction (SAED), and energy dispersive x-ray spectroscopy (EDS) observations were performed using a JEM-2010 (JEOL, Kyoto, Japan) transmission electron microscope operating at an accelerating voltage of 200 kV. GaSe flakes prepared by ultrasonic exfoliation in ethanol were transferred onto a TEM grid through direct transfer. To minimize oxidation of GaSe under ambient conditions, the transfer should be completed quickly.

For temperature-dependent SE measurements, a sample surface containing the c- and a (or b)-axes was prepared by polishing with a 0.05-μm colloidal silica suspension applied on a polishing cloth. The sample was then loaded into a cryostat and evacuated using a turbomolecular pump with a base pressure of approximately 10^−8^ Torr. Pseudodielectric function values were obtained from 0.74 to 6.42 eV at temperatures of 27 K, with a 25 K interval from 50 to 300 K, using a rotating-compensator SE (J.A. Woollam Inc., RC2 model, Lincoln, Nebraska, USA) at the Multi-dimension Material Convergence Research Center of Kyung Hee University at an angle of incidence (AOI) of 68.80 degrees.

## 3. Results and Discussion

### 3.1. Structural Characteristics

Figure 1c shows the XRD pattern of the GaSe single crystal in the c-plane, in which diffracted peaks at 11.14, 22.34, 33.77, 45.50, and 57.80° corresponding to (002), (004), (006), (008), and (020) planes were clearly observed. This result is well in agreement with previous reports of the hexagonal structure of GaSe [52,53]. To further explore the crystal quality of this material, HRTEM is employed for high-resolution imaging. Figure 1d shows the HRTEM image of the c-plane of the GaSe structures with the d-spacing (100) lattice plane of 3.2 nm. The SAED pattern confirms the six-fold rotational symmetry of the hexagonal structure of single-crystal GaSe. The stoichiometry of the crystals is determined by the energy-dispersive X-ray spectroscopy (EDS) analysis, as shown in Appendix A. Since we used a TEM grid made of copper with several layers of graphene, the spectrum shows the copper and carbon peaks.

GaSe has a hexagonal structure with a space group $D_{6h}^{4}$; thus, 24 normal modes of vibration at the Γ-point of the Brillouin zone are presented as [18,19]
(1)$$\Gamma =2A_{1g}+2A_{2u}+2B_{2g}+2B_{1u}+2E_{1g}+2E_{1u}+2E_{2g}+2E_{2u}$$
of which there are six Raman active modes. The pertinent Raman tensors expressed in the hexagonal crystal principal axes for the $A_{1g}$ modes can be described as below [37,54]
(2)$$A_{1g}:\;\;\;\;\left[\begin{array}{ccc} a & 0 & 0\\ 0 & a & 0\\ 0 & 0 & b \end{array}\right]$$ The intensity dependence of $A_{1g}$ now is
(3)$$I(A_{1g}^{1})\propto {\left[a\cos^{2}\theta +b\sin^{2}\theta \right]}^{2}$$
where θ is the polarization angle relative to the c-axis. Figure 2a shows the Raman spectra of single crystal GaSe on the in-plane (black curve) and out-of-plane with different polarized angles. Here, $0^{o}$ denotes the polarization of the incident where polarized light is parallel to the c-axis. In the (ab) plane, the $A_{1g}^{1}$ and $A_{1g}^{2}$ peaks at 132.96 and 306.45 cm^−1^, respectively, and the $E_{2g}^{1}$ peak at 211.45 cm^−1^ are clearly seen, while the $E_{1g}^{2}$ peak is mostly absent. However, the $E_{1g}^{2}$ peak can be clearly observed when measurements are conducted out-of-plane. It is interesting that the relative intensity of the $A_{1g}^{1}$ peak to others is maximum when the polarization of incident light is parallel to the c-axis. Appendix A shows further detail of the Raman intensity depending on the polarization angles. Figure 2b shows the dependence of $A_{1g}^{1}$ measured on the out-of-plane to the polarization of incident light fitted by Equation (2). This result confirms the two-fold rotational symmetry of the out-of-plane.

### 3.2. Critical Point (CP) Analysis

In order to remove the contribution of surface roughness to the intrinsic dielectric function of GaSe single crystal, point-by-point fitting has proceeded (Appendix A). Figure 3a,b show the dielectric function of single-crystal GaSe along the c- and a(or b)- axes, respectively, from 27 to 300 K, with offset by increments of 2 relative to the 27 K spectrum for the c-axis and 5 for the a(or b)-axis. The blue shift and sharpening of CPs with temperature decreasing are clearly seen. There are 8 CPs observed in the c-axis and marks as $E_{exc}^{c}$ for exciton and $E_{0}^{c}$ to $E_{6}^{c}$ for others as shown in Figure 3a. For the a-axis, only six CPs are detected from 1 to 6 eV and denoted by $E_{1}^{a}$ to $E_{6}^{a}$, as shown in Figure 3b. The anisotropy in the out-of-plane of single crystal GaSe is clearly presented by the difference in bandgap and lineshapes along the c- and a-axes. The direct bandgap in the c-axis is about 2.1 eV and dominance the of excitonic feature is observed. This result is similar to previous work [22,24,25,26,27,32]. In the a-axis, the bandgap is up to 3.2 eV and intensity is significantly higher than in the c-axis. This result is in good agreement with previous SE measurements by Choi et al. [29]. As mentioned above, Meyer et al. [30] published the dielectric functions of a GaSe single crystal along the principal axes. It is worth comparing their results with current work, as shown in Appendix A.

To enhance the resolution of the overlapping CP structure, the second derivatives $d^{2}\varepsilon /dE^{2}$ were performed numerically. Linear interpolation and Gauss–Hermite filtering were applied for noise reduction with minimal lineshape distortion [55,56]. The standard analytic CP expression then was fitted to extract CP parameters
(4)$$\frac{d\varepsilon^{2}}{dE^{2}}=\left\{\begin{array}{c} n\left(n-1\right)Ae^{i\phi }{\left(ћ\omega -E+i\Gamma \right)}^{n-2},\;\;n\neq 0\\ Ae^{i\phi }{\left(ћ\omega -E+i\Gamma \right)}^{-2},\;\;n=0 \end{array}\right.,$$
where A, ϕ, E, and Γ are the amplitude, phase, energy, and broadening of CP, respectively. The exponent *n* has values of −1, −1/2, 0, and 1/2 corresponding to excitonic, one-, two-, and three-dimensional CPs. Both real and imaginary parts of ε are fitted simultaneously. All CPs show the best results with the excitonic lineshape (n=−1) except for the $E_{0}^{c}$ and $E_{1}^{c}$ structures. This is similar to the CP analysis of the a-axis data reported in ref. [29].

Figure 4a,b show the second derivative and their best fits of ε at 27 K along the c- and a-axes, respectively. The data are open circles, while the best fits of $d^{2}\varepsilon_{1}/dE^{2}$ and $d^{2}\varepsilon_{2}/dE^{2}$ correspond to the solid and dashed lines. For clarity, the number of data points was approximately reduced; only the data for $d^{2}\varepsilon_{1}/dE^{2}$ are shown and the data from 3.5 to 6.0 eV are multiplied by three as shown in Figure 4a. The existence of both exciton $E_{exc}^{c}$ and fundamental bandgap $E_{0}^{c}$ are clearly identified in the second derivative spectrum. We note that this observation of the separation and binding energy of excitons may play an important role in clarifying the operation and efficiency of nanodevices. See Appendix A for a comparison between single and two CPs of the fitting.

### 3.3. Identification of CPs

Figure 5 shows the energy band structure obtained using the mBJ method for bandgap correction. Appendix A shows the energy and structure of the density of state obtained using and not using the mBJ method. As indicated in previous studies [49,57,58], the difference between band structures with and without spin-orbit coupling (SOC) is small enough to be negligible in the current analysis of low-energy bandgaps; hence, non-SOC computation is conducted here. This is plotted with partial orbitals, where blue, brown, green, and red represent the orbitals *s*, $p_{x}$, $p_{y}$, and $p_{z}$, respectively. As shown in the energy band structure, the top of the valance band (VB) and the bottom of the conduction band (CB) are localized at the Γ-point; therefore, the direct optical transition is allowed in this situation, as marked by $E_{0}^{c}$.

Figure 5a shows the band structure with the distribution of the orbitals *s* and $p_{z}$, which are considered to be the main contributors to band-to-band transitions of the c-axis. The fundamental direct bandgap $E_{0}^{c}$ occurs between the first VB and the first CB at the Γ-point corresponding to the contribution of $p_{z}$ (at VB) and s (at CB) orbitals; this leads to the dominance of $E_{0}^{c}$ in the c-axis while it is absent in the a(or b)-axis. The fact that the bandgap transitions of GaSe occur between the *p_z_* and *s* orbitals should provide valuable information for characterizing excitonic behavior in this material. The calculated result indicates that the energy of this CP is about 2.29 eV. This value is in good agreement with the experimental data extracted by the second derivative analysis, which estimates 2.108 eV at 27 K as listed in Table 1. As considered before, Meyer et al. [30] were the first to report the dielectric functions of the two principal axes of GaSe; however, no CP values were obtained for comparison. The binding energy is lower than the ground state of 27 meV (at 27 K) similar to those of previous reports [22,24,59,60]. These other CPs are noted by arrows from $E_{1}^{c}$ to $E_{6}^{c}$ as shown in Figure 5a and their values are listed in Table 2.

Figure 5b indicates the band-to-band transitions of CPs along the a(or b)-axis and denoted from $E_{1}^{a}$ to $E_{6}^{a}$, in which all CPs are assigned from the main contribution of *s*, $p_{x}$, and $p_{y}$ orbitals. The theoretical values of CP energies are well consistent with the SE data as listed in Table 1. The previously reported results [28,29,61] are also comparable to our theoretical and experimental values, which confirm the validity of our work.

**Table 1 nanomaterials-14-00839-t001:** CP energies (eV) at 27 K and room temperature (RT) compared to previously reported data and band calculations.

Axes	CPs	This Work		References						This Work
		SE		SE ^a^	SE ^b^	PL ^c^	PL ^d^	PL ^e^	MWS ^f^	DFT
		27 K	300 K	RT	RT	6 K ^c^	RT	80 K	300 K	
c-axis	$E_{exc}^{c}$	2.081	1.967	_	_	2.11	2.00	2.098	_	_
	$E_{0}^{c}$	2.108	1.997	_	_	_	_	_	_	2.29
	$E_{1}^{c}$	3.89	3.77	_	_	_	_	_	_	3.94
	$E_{2}^{c}$	4.06	4.03	_	_	_	_	_	_	4.12
	$E_{3}^{c}$	4.65	4.53	_	_	_	_	_	_	4.72
	$E_{4}^{c}$	4.78	4.75	_	_	_	_	_	_	4.83
	$E_{5}^{c}$	5.32	5.26	_	_	_	_	_	_	5.39
	$E_{6}^{c}$	5.95	5.91	_	_	_	_	_	_	5.90
a-axis	$E_{1}^{a}$	3.36	3.18	3.23	3.23	_	_	_	3.22	3.30
	$E_{2}^{a}$	3.78	3.66	3.67	3.75	_	_	_	3.68	3.79
	$E_{3}^{a}$	4.80	4.68	4.60	4.69	_	_	_	4.78	4.82
	$E_{4}^{a}$	5.04	4.93	4.80	5.02	_	_	_	5.07	5.00
	$E_{5}^{a}$	5.34	5.23	_	5.45	_	_	_	5.48	5.50
	$E_{6}^{a}$	5.76	5.74	_	5.72	_	_	_	5.75	5.79

^a^ Ref. [28], ^b^ Ref. [29], ^c^ Ref. [24], ^d^ Ref. [62], ^e^ Ref. [59], ^f^ Ref. [61] modulation wavelength spectra (MWS).

Figure 6 shows the dependence of CP energies on temperature from 27 to 300 K. The open dots are results of the second-derivative analysis and the solid lines are the best fits obtained by a phenomenological expression that contains the Bose–Einstein statistical factor for phonons [63,64]:(5)$$E(T)=E_{B}-a_{B}\left(1+\frac{2}{e^{\Theta /T}-1}\right),$$
where Θ is the mean frequency of the phonons and $a_{B}$ is the interaction strength between electrons and phonons. The temperature dependence of the $E_{5}^{c}$ to $E_{7}^{c}$ in the c-axis and $E_{5}^{a}$ to $E_{6}^{a}$ in the a-axis are described by the linear equation
(6)$$E(T)=E_{L}-\lambda T,$$
where λ is the temperature coefficient and dE/dT is an adjustable parameter along with $E_{L}$. The best-fit parameters of Equations (5) and (6) for these CPs are listed in Table 2. We obtained the mean phonon frequencies and interaction strengths of $E_{exc}^{c}$ and $E_{0}^{c}$ peaks, which have similar values. This observation indicates that both transitions originate from the same origin, providing important insights into the excitonic properties.

**Table 2 nanomaterials-14-00839-t002:** The best-fitting parameters of the temperature dependences of the CPs of single crystal GaSe along the c- and a-axes.

Axes	CPs	$E_{B}$ (eV)	$a_{B}$ (meV)	Θ (K)	$E_{L}$ (eV)	λ (10^−4^ eVK^−1^)
c-axis	$E_{exc}^{c}$	2.15 ± 0.01	65 ± 6	232 ± 15	_	_
	$E_{0}^{c}$	2.17 ± 0.01	67 ± 6	238 ± 15	_	_
	$E_{1}^{c}$	3.96 ± 0.02	72 ± 19	238 ± 44	_	_
	$E_{2}^{c}$	4.09 ± 0.02	37 ± 22	419 ± 133	_	_
	$E_{3}^{c}$	4.77 ± 0.03	117 ± 36	345 ± 63	_	_
	$E_{4}^{c}$	_	_	_	4.79 ± 0.01	1.15 ± 0.08
	$E_{5}^{c}$	_	_	_	5.32 ± 0.01	1.85 ± 0.15
	$E_{6}^{c}$	_	_	_	5.95 ± 0.01	1.99 ± 0.40
a-axis	$E_{1}^{a}$	3.44 ± 0.03	86 ± 30	209 ± 55	_	_
	$E_{2}^{a}$	3.84 ± 0.01	59 ± 3	209 ± 10	_	_
	$E_{3}^{a}$	4.84 ± 0.01	37 ± 8	139 ± 24	_	_
	$E_{4}^{a}$	5.15 ± 0.02	104 ± 22	309 ± 41	_	_
	$E_{5}^{a}$	_	_	_	5.35 ± 0.01	3.69 ± 0.32
	$E_{6}^{a}$	_	_	_	5.76 ± 0.01	0.62 ± 0.24

## 4. Conclusions

In summary, this study presents the anisotropic dielectric responses of GaSe, a uniaxial crystal, along its principal axes. These responses were measured across a wide range of energy, from 0.74 to 6.42 eV, and at various temperatures ranging from 27 K to 300 K. The results show that the fundamental bandgap and exciton are only observed in the c-direction. These observations are consistent with the first-principles calculation, which confirms that the band-to-band transition of *s* and *p_z_* orbitals at the Γ-point is the main contributing factor. The temperature dependences of CP energies were determined using either a linear equation or a phenomenological expression incorporating the Bose–Einstein statistical factor. These showed a blue shift and enhanced structure at low temperatures as a result of reduced lattice constant and electron–phonon interactions. This study expands our understanding of the optical characteristics of GaSe and can provide valuable insights into the precise engineering of optoelectronic devices.

## Figures and Tables

**Figure 1 nanomaterials-14-00839-f001:**
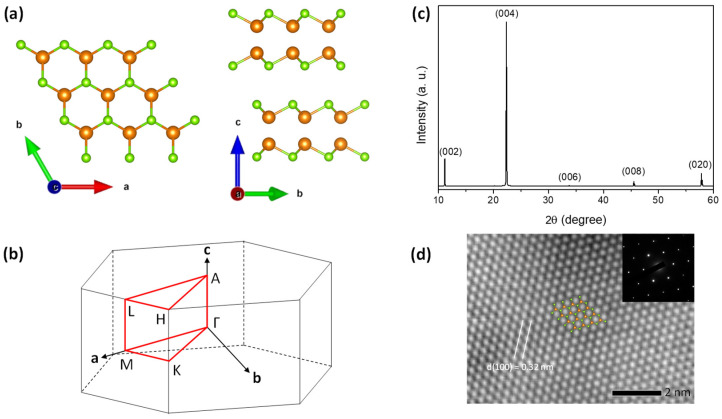
(**a**) The hexagonal β-GaSe crystal structure. The Ga atoms are orange and the Se atoms are green. (**b**) Brillouin zone of the GaSe structure. (**c**) XRD pattern on a GaSe single crystal aligned along the (001) plane. (**d**) HR-TEM image of the sample in the c-plane and the inset shows the SAED pattern.

**Figure 2 nanomaterials-14-00839-f002:**
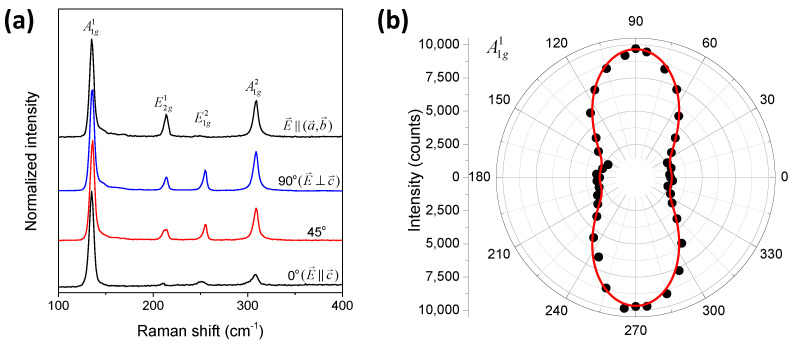
(**a**) Top-down: polarized Raman spectra of the GaSe single crystal on the in-plane and out-of-plane. (**b**) $A_{1g}^{1}$-peak Raman intensity as a function of the angle and the red line depicts the fitting result with the equation. Details are given in the text.

**Figure 3 nanomaterials-14-00839-f003:**
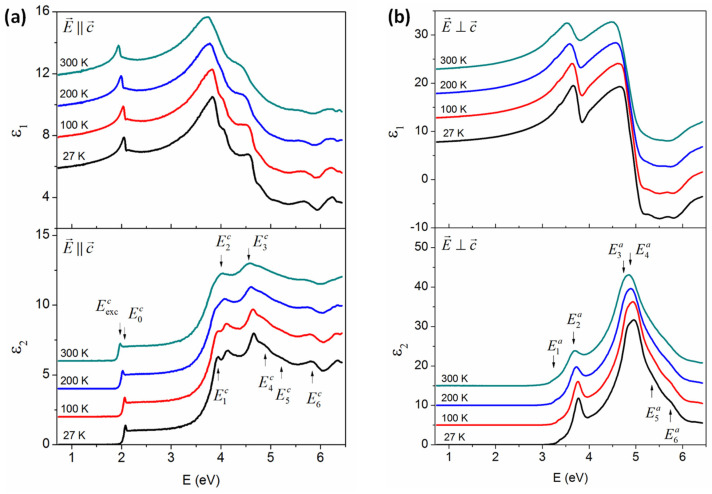
Real $\left(\varepsilon_{1}\right)$ and imaginary $\left(\varepsilon_{2}\right)$ parts of the dielectric function of single-crystal GaSe along the (**a**) c- and (**b**) a-axes. The spectra are offset by increments of 2 and 5 relative to the 27 K spectrum in the c- and a-axes, respectively.

**Figure 4 nanomaterials-14-00839-f004:**
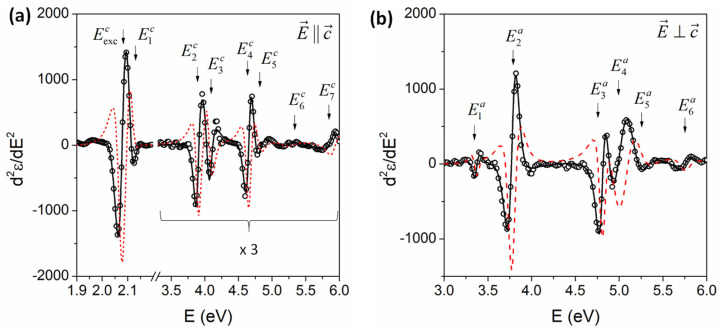
Second derivative of ε with respect to the energy of GaSe at 27 K along the (**a**) c-axis and (**b**) a-axis. The solid and dashed curves are the best fits of the real and imaginary parts of ε, respectively. For clarity, the data for the imaginary parts are not shown.

**Figure 5 nanomaterials-14-00839-f005:**
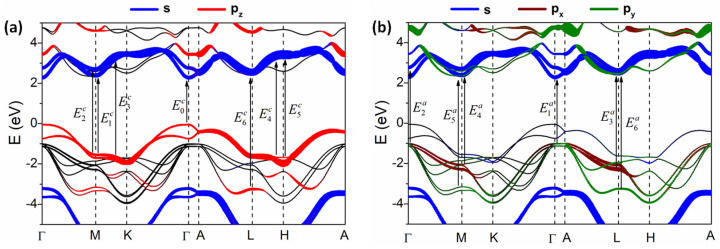
(**a**,**b**) Band structure of GaSe calculated using the PBE method for bandgap correction. Partial orbitals of atoms are presented for each energy band. Blue, brown, green, and red represent *s*, *p_x_*, *p_y_*, and *p_z_*, respectively. Arrows denote the main transitions associated with the CPs.

**Figure 6 nanomaterials-14-00839-f006:**
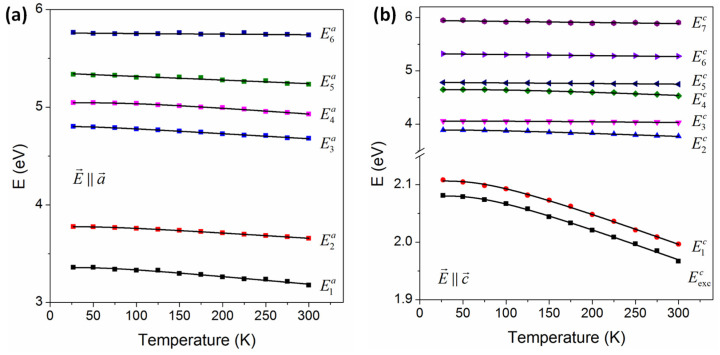
Temperature dependences of the CP energies (open symbols) of GaSe and the best fits (solid lines) for CPs of the (**a**) c- and (**b**) a-axes.

## Data Availability

The data presented in this study are available on request from the corresponding author.

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
