# Peer review of "A Systematic Study of the Temperature Dependence of the Dielectric Function of GaSe Uniaxial Crystals from 27 to 300 K"

_nanomaterials, 2024, doi:10.3390/nano14100839_

Round 1

Reviewer 1 Report

Comments and Suggestions for Authors

In the manuscript entitled “A systematic study of the temperature dependence of the dielectric function of GaSe uniaxial crystals from 27 to 300 K”, the authors investigated the temperature dependence and the anisotropic behavior of the dielectric function of the GaSe. Via experimental and numerical techniques, they showed features such as the blue-shift and sharpening of CPs as decreasing temperature, and the band-to-band transition induced exciton feature and fundamental bandgap. The manuscript is well written and the data are solid. However, to be considered publication, I have two main issues:

Firstly, in the introduction, the authors briefly reviewed on the optical properties of Gallium selenide, they dated back to the 1970s but with no update covered recent decade. The authors should comment on this research gap.

This also brings to my second point. The authors are required to show the scientific significance of their finding, i.e. (a) the significance of “The existence of both exciton E^c_exc and fundamental bandgap E^c_0 (line 218)”, and (b) the underlying mechanisms for band-to-band transition accounting for the “exciton feature and fundamental bandgap”, as well as (c) the novelty in the presence manuscript.

Other minor comments, (a) Line 28, “fundamental”, (b)Caption of figure S4, the word “fitsitting” should be checked. (c) line 55, the citation must be addressed.

Author Response

Dear Editors and Reviewers:

Thank you for your letter and for the reviewers’ comments concerning our manuscript entitled “A systematic study of the temperature dependence of the dielectric function of GaSe uniaxial crystals from 27 to 300 K”. Recently according to the suggestion of editors and reviewers, we have carefully studied the comments and made corrections, hoping to be approved. Any revisions to the manuscript were marked up with "yellow highlight" form. The main corrections in this article and responses to reviewer comments are as follows:

Reviewer 1

In the manuscript entitled “A systematic study of the temperature dependence of the dielectric function of GaSe uniaxial crystals from 27 to 300 K”, the authors investigated the temperature dependence and the anisotropic behavior of the dielectric function of the GaSe. Via experimental and numerical techniques, they showed features such as the blue-shift and sharpening of CPs as decreasing temperature, and the band-to-band transition induced exciton feature and fundamental bandgap. The manuscript is well written and the data are solid. However, to be considered publication, I have two main issues:

 We sincerely appreciate the comments of the reviewer. We tried our best to revise our manuscript according to reviewer’s comments as below.

Firstly, in the introduction, the authors briefly reviewed on the optical properties of Gallium selenide, they dated back to the 1970s but with no update covered recent decade. The authors should comment on this research gap.

Answer: We found that we made a mistake to focus only on recent ellipsometry work, so we did not include optical studies done by other methods. Therefore, we updated recent optical investigation of GaSe, and we added those in the text from lines 58 to 66: “In 2008, Cui et al. [33] reported the photoluminescence (PL) of GaSe and GaSe:In at 9 K. They discovered that the peak of the exciton bound to the acceptor disappeared and the peak of the donor-acceptor pair appeared in the GaSe crystal after indium doping. Zhang et al. [34] studied the temperature-dependent PL emission from unstrained and strained GaSe nanosheets and found that the formation of new peaks in strained samples can be attributed to the recombination of bound excitons. In 2022, Usman et al. [23] investigated the thickness and temperature dependencies of PL in few-layer GaSe. They observed that the PL intensity linearly increases with the number of layers, while the peak position increases as the layer number decreases.”

This also brings to my second point. The authors are required to show the scientific significance of their finding, i.e. (a) the significance of “The existence of both exciton E^c_exc and fundamental bandgap E^c_0 (line 218)”, and (b) the underlying mechanisms for band-to-band transition accounting for the “exciton feature and fundamental bandgap”, as well as (c) the novelty in the presence manuscript.

Answer:

(a) Exciton and fundamental bandgap are intrinsic and crucial physical properties of semiconductors. To comprehend the behavior of semiconductor materials, it is essential to have a clear understanding of exciton characteristics. Excitonic properties frequently play a crucial role in determining the potential applications of these materials in optoelectronic devices, such as photovoltaics, light-emitting diodes, and even lasers.

(b) The mechanism to identify the band-to-band transitions of the fundamental bandgap of GaSe is well presented in the section "3.3. Identification of CPs". In this section, we provide the energy band structure with the distribution of the orbitals s, px, py, and pz to explain the anisotropic properties of this material. This method is widely known and has been reported for many interesting anisotropic materials.

(c) As shown in the manuscript, only a few works have studied the anisotropic optical properties of GaSe. For SE measurements of GaSe, only three studies have been reported so far, excluding our own. The first SE study was conducted by Meyer in 1973, but the obtained results for the two principal axes were relatively crude. Meanwhile, Choi et al. (2009) and Isik et al. (2016) measured the dielectric function of GaSe on the cleavage plane only. To emphasize the significant contribution of our current work, we added Figure S4 which compares Meyer’s result with our own. We added sentences “As mentioned above, Meyer et al. [30] published the dielectric functions of GaSe single crystal along the principal axes. It is worth to compare their results with current work, as shown in Figure S4.” in line 208

We really appreciate the invaluable comments provided by the reviewer to improve the quality of our work. In response, we have included several additional sentences in the text to clarify the points mentioned by the reviewer. These sentences are as follows: “We note that this observation of the separation and binding energy of excitons may play important role in clarifying the operation and efficiency of nanodevices.” in line 233. “The fact that the bandgap transitions of GaSe occurs between the pz and s orbitals should provide valuable information for characterizing excitonic behavior in this material.” in line 255. “We obtained the mean phonon frequencies and interaction strengths of Ecexc and Ec0 peaks have similar values. This observation indicates that both transitions originate from the same origin, providing important insights into the excitonic properties.” in line 294.

Other minor comments, (a) Line 28, “fundamental”, (b)Caption of figure S4, the word “fitsitting” should be checked. (c) line 55, the citation must be addressed.

Answer: We sincerely appreciate correcting our mistake.

We revised as follow.

(a) In line 28: We changed “fundmental” to fundamental”.

(b) We changed “fitsitting” to “fitting”.

(c) In line 57: We added “[32]”

Additional changes:

In line 29: We changed “s pz orbitals” to “s and pz orbitals”

In line 283: We miswrote “Figure 5”. We changed it to “Figure 6”

In line 216: We changed “- second” to “second”

Reviewer 2 Report

Comments and Suggestions for Authors

The paper presents a study of the dielectric function of GaSe over a wide range of temperatures. It provides an expansion on existing work. The paper is well written and the results presented underpin the conclusions that have been drawn. In all this is a good paper that requires minimal corrections.

The major question that arises from the results presented here, is the EDX data as presented in the supplementary information. Unsurprisingly for a single crystal of GaSe those two atoms are present, but there is also an equal amount of copper present in the samples. Where does this copper come from?

Minor corrections:

- Line 57: the authors are referring to the work by Antonioli et al. but there is no reference.

- Line 160: has a hexagonal instead of has hexagonal.

- Line 204: performed numerically rather than performednumerically.

- Line 209: the font of n appears odd

- Line 217: it should be is rather than isare.

- Figure 4: the fontsize inside the figure, especially the sub- and superscript is impossible to read.

- Line 267: The reference to Figure 5 should probably be to Figure 6.

Comments on the Quality of English Language

The quality of the English is fine, some minor issues were detected and listed in the general comment section.

Author Response

Dear Editors and Reviewers:

Thank you for your letter and for the reviewers’ comments concerning our manuscript entitled “A systematic study of the temperature dependence of the dielectric function of GaSe uniaxial crystals from 27 to 300 K”. Recently according to the suggestion of editors and reviewers, we have carefully studied the comments and made corrections, hoping to be approved. Any revisions to the manuscript were marked up with "yellow highlight" form. The main corrections in this article and responses to reviewer comments are as follows:

Reviewer 2

The paper presents a study of the dielectric function of GaSe over a wide range of temperatures. It provides an expansion on existing work. The paper is well written and the results presented underpin the conclusions that have been drawn. In all this is a good paper that requires minimal corrections.

Answer: We sincerely appreciate the positive comments from the reviewer.

The major question that arises from the results presented here, is the EDX data as presented in the supplementary information. Unsurprisingly for a single crystal of GaSe those two atoms are present, but there is also an equal amount of copper present in the samples. Where does this copper come from?

Answer: The TEM grid we used was made of copper with several layers of graphene deposited on top. This is why the EDX spectrum showed both Cu and C elements. So we added in the line 163 of the main text to explain this: “Since we used TEM grid made of copper with several layers of graphene, the spectrum shows the copper and carbon peaks.”

Minor corrections:

- Line 57: the authors are referring to the work by Antonioli et al. but there is no reference.

- Line 160: has a hexagonal instead of has hexagonal.

- Line 204: performed numerically rather than performednumerically.

- Line 209: the font of n appears odd

- Line 217: it should be is rather than isare.

- Figure 4: the fontsize inside the figure, especially the sub- and superscript is impossible to read.

- Line 267: The reference to Figure 5 should probably be to Figure 6.

Answer: All our mistakes pointed by the reviewer were corrected. We really thank for these comments.

Round 2

Reviewer 1 Report

Comments and Suggestions for Authors

I think the manuscript is ready for publication.

A minor typos line 207, "in the c-axis" .